# Local and Landscape Factors Influence Plant-Pollinator Networks and Bee Foraging Behavior across an Urban Corridor

Gabriella L. Pardee [1,*], Kimberly M. Ballare [1,2], John L. Neff [3], Lauren Q. Do [1], DianaJoyce Ojeda [1], Elisa J. Bienenstock [4,†], Berry J. Brosi [5,†], Tony H. Grubesic [6,†], Jennifer A. Miller [7,†], Daoqin Tong [8,†] and Shalene Jha [1,9]

1   Department of Integrative Biology, University of Texas at Austin, Austin, TX 78712, USA
2   Ecology and Evolutionary Biology Department, University of California Santa Cruz, Santa Cruz, CA 95064, USA
3   Central Texas Melittological Institute, Austin, TX 78731, USA
4   Watts School of Public Service and Community Solutions, Arizona State University, Tempe, AZ 85281, USA
5   Department of Biology, University of Washington, Seattle, WA 98195, USA
6   Center for Geospatial Sciences, University of California Riverside, Riverside, CA 92521, USA
7   Department of Geography and the Environment, University of Texas at Austin, Austin, TX 78712, USA
8   School of Geographical Sciences and Urban Planning, Arizona State University, Tempe, AZ 85281, USA
9   Lady Bird Johnson Wildflower Center, Austin, TX 78739, USA
*   Correspondence: gabriellapardee@gmail.com
†   These authors contributed equally to this work.

**Abstract:** Given widespread concerns over human-mediated bee declines in abundance and species richness, conservation efforts are increasingly focused on maintaining natural habitats to support bee diversity in otherwise resource-poor environments. However, natural habitat patches can vary in composition, impacting landscape-level heterogeneity and affecting plant-pollinator interactions. Plant-pollinator networks, especially those based on pollen loads, can provide valuable insight into mutualistic relationships, such as revealing the degree of pollination specialization in a community; yet, local and landscape drivers of these network indices remain understudied within urbanizing landscapes. Beyond networks, analyzing pollen collection can reveal key information about species-level pollen preferences, providing plant restoration information for urban ecosystems. Through bee collection, vegetation surveys, and pollen load identification across ~350 km of urban habitat, we studied the impact of local and landscape-level management on plant-pollinator networks. We also quantified pollinator preferences for plants within urban grasslands. Bees exhibited higher foraging specialization with increasing habitat heterogeneity and visited fewer flowering species (decreased generality) with increasing semi-natural habitat cover. We also found strong pollinator species-specific flower foraging preferences, particularly for Asteraceae plants. We posit that maintaining native forbs and supporting landscape-level natural habitat cover and heterogeneity can provide pollinators with critical food resources across urbanizing ecosystems.

**Keywords:** bee communities; pollination; pollinator generality; pollen preference; semi-natural habitat; specialization

## 1. Introduction

The urban-to-rural interface has long been shown to drive changes in animal community composition via shifts in food and nesting resources, alteration in biotic and abiotic factors, and the impediment of an organism's ability to disperse into hospitable habitat [1,2]. Despite this fact, researchers have only begun examining how changes in habitat composition affect mutualistic relationships across human-modified landscapes [3,4]. This effort is complicated because natural habitats within urban areas are often highly heterogenous and thus vary in their degree of quality and connectivity, which can directly

affect mutualistic interactions [5,6]. For example, past work has shown that increases in landscape-level natural habitat cover can result in higher pollination services [7] and that increases in landscape-level heterogeneity can lead to increases in mutualistic plant-ant co-occurrences [8]. Furthermore, given that human-dominated habitats, which we describe as urbanizing environments that have high population densities, are expanding in size and number due to human population growth [9], it is essential to deepen our understanding of how landscape-level habitat composition impacts important mutualistic relationships in the face of these continued global changes.

The mutualistic interaction between plants and pollinators is particularly critical in the context of global change because pollinators contribute more than 300 bn USD worldwide in ecosystem service provisioning [10], given that the majority of plant species rely on pollinators for successful reproduction [11,12]. Previous studies have shown that both local and landscape characteristics of urban parks and gardens can affect the richness and abundance of bee species [13–15], with potential subsequent effects on plant-pollinator interactions and pollination success. Indeed, plant-pollinator network indices have successfully predicted community stability and ecosystem function [16], particularly within human-dominated landscapes [17,18]. For example, past work has indicated that habitat fragmentation due to increased agriculture alters network structure and size, which can lead to increased extinction rates, disruption of ecosystem functioning, and functional homogenization of pollinator communities [19,20]. Further, in simplified habitats with limited resources for bees, plant-pollinator networks are often less specialized, meaning that pollinators are interacting with a larger group of plant species than in more complex habitats [21]. In contrast, a more heterogeneous habitat may allow pollinators to seek out specific flowers, leading to more specialized plant-pollinator networks [22]. Further, network indices such as foraging specialization and resource partitioning are predictive of many pollination success metrics, such as stigmatic pollen deposition [17], highlighting the high ecological value of these network quantifications.

In addition to leveraging plant-pollinator interactions for predicting function and community stability, it is essential to understand pollinator species-specific plant preferences in order to effectively restore optimal pollinator habitat. Animal foraging preferences can be beneficial to consider when designing conservation habitat [23] and are classically defined as the resources that are utilized in proportionally higher amounts than are available across a landscape [24]. For many pollinators, pollen is the fundamental food source for larval development and determines adult reproductive success [25]. As pollen is highly variable in its nutritional content across species [26], many pollinators seek out flower species that are most beneficial for their offspring [27]. Despite the value of understanding pollen preference for pollinator conservation, past network interaction research has often focused on pollinator visitation data (not pollen collection), irrespective of whether insects actually collected pollen [28]. Thus, assessment of pollen floral preference is often missing from pollinator community ecology research, even though it could provide key information for improving vegetation management efforts.

In this study, we quantify pollen-based plant-pollinator networks and pollinator species-specific pollen usage across a ~350 km rapidly urbanizing linear corridor in the southern US to determine the effects of land use change on plant-pollinator interactions and pollinator foraging preferences. We focused our sampling on bees as they are among the most effective pollinators [29,30], especially in the southern U.S. [31]. We investigated if local and landscape habitat features impact key plant-pollinator network characteristics, specifically specialization, connectance, nestedness, and pollinator generality, and quantified whether bees show pollen preferences within urban grassland habitats. We hypothesized that network foraging specialization would increase and pollinator generality would decrease with an increased amount of surrounding semi-natural habitat, indicative of more selective foraging behavior, as seen in past studies [32]. In contrast, network connectance and nestedness would increase with increased floral abundance and vegetation cover, as these characteristics indicate greater habitat complexity and food resources [33,34].

We also hypothesized that native pollinators would exhibit preferences for both native and non-native plant species, likely those within Asteraceae, as seen in other pollen preference studies conducted in Texas, U.S. [35]

## 2. Methods

### 2.1. Study System

We conducted this study across ~350 km of the rapidly urbanizing I-35 highway corridor in Central Texas, USA. This corridor, which encompasses five of the twenty fastest-growing large cities in the U.S. [36], cuts through highly fragmented savannah and grassland habitat [37]. Both the Austin and Dallas-Fort Worth metroplexes expanded by more than 4% per year from 2000 to 2010 [38], highlighting the urgent need for urban ecological research across these regions. We selected 20 grassland sites within this linear corridor (10 across each metroplex) based on past work in the Central Texas region [13]. These sites are all managed as wildlife refuges or city/state parks and are dominated by native grassland vegetation. All sites have minimal to no active disturbance regimes (e.g., fire or grazing), except for infrequent mowing in some sites that occurred less than two times per year and outside of the sampling season. Sites were located along a gradient of developed land and were separated by at least 2 km, as this captures typical maximum foraging distances for bees [39]. In each site, we established a 50 × 50 m plot to sample local habitat characteristics as well as bee and flowering plant communities. We visited each site three times during the late spring and early summer of 2013, specifically in the first two weeks of May, June, and July, as described in [13]. We chose to sample during the late spring and early summer as this period has both high pollinator abundance and low year-to-year variation in rainfall and is thus likely to produce the most representative sample of the typical bee and flowering community [13].

### 2.2. Local and Landscape Characteristics

Within each 50 × 50 m sampling plot, we laid out three 50 m transects, running North to South, at 15, 25, and 35 m to measure the flowering plant community and local habitat characteristics, as described in Ballare et al. [11]. Within each quadrat, we recorded the number of flowering forb species, the number of inflorescences per flowering species, and the percent cover of live ground vegetation, bare ground, rocky/impervious surface, and leaf litter. Previous studies have shown these factors to be important resources for bee nesting and foraging [15,40]. Overall, sites had a mean of 53.29% ground vegetation, 12.94% bare ground, 2.78% rocky/impervious surface, and 26.60% leaf litter, as well as a mean of 402.55 flowering inflorescences, and 5.98 flowering species.

We used the 2011 National Land Cover Database (NLCD; 30 m resolution and minimum mapping areas of five pixels: [41]) dataset to quantify the amount of semi-natural, agricultural, and developed land within a 2 km buffer of each site and calculated these values in qGIS v. 2.14 [13,42,43]. Specifically, we summed the percentage area of each of 11 land use types in 2 km buffers surrounding each site into three land cover categories representing broad differences in bee nesting and foraging habitat (as per Ritchie et al., 2016 [35]): (1) developed land, comprised of high intensity developed, medium intensity developed, low intensity developed, and developed open space (NLCD categories 21–24); (2) pasture and cropland, comprised of pasture/hay and agriculture crops (NLCD categories 81–82); and (3) semi-natural land, comprised of deciduous forest, evergreen forest, mixed forest, grassland/herbaceous, and shrub/scrub (NLCD categories 41–43, 71, 52). Finally, as habitat heterogeneity could also play a role in bee nesting and foraging behavior, we quantified heterogeneity as the total number of land use patches of all 15 possible NLCD land use types within each 2 km buffer, calculated using qGIS, as in [13]. Sites were surrounded by 0–100% developed land, 0–63% cropland, and 0–98% semi-natural habitat within a 2 km radius, with a mean of 10 land-use types within a 2 km radius.

### 2.3. Pollinator Sampling

We focused on wild native bees in our study, and thus did not collect non-native managed honey bees (*Apis mellifera*) as none of our sites contained hives and therefore had low abundances of honey bees. a. We sampled bees systematically within our 50 × 50 m plot by collecting any native bee seen actively foraging on a flower using hand nets, as done in previous pollen studies examining floral preferences [44–47]. To reduce observation bias, two trained researchers each sampled half of the plot for 15 min each, then switched sides for the remaining 15 min, for a total netting time of 1 h per site in each of three sampling rounds (May, June, and July), as in [48]. To reduce cross-contamination, each bee was placed into a separate vial upon collection, and the observer shook out the hand net to remove any pollen. All specimens were pinned, labeled with their site-round and floral association, and identified to the lowest taxonomic level. Given temporal proximity in the sampling round, we combined interactions across the three rounds as in past network studies (i.e., [20]) for a total of 20 networks.

### 2.4. Pollen Identification

All bees were washed for pollen load analysis by gently vortexing the entire specimen in 5.5 mL of ethanol to maximize pollen extraction, centrifuging the tube to isolate the pollen pellet, and removing 5 mL of the ethanol supernatant, as described in [35]. To create the pollen slide for each individual bee, 20 microliters of the vortexed pollen pellet, plus ethanol solution, was added to 60 microliters of fuchsin + glycerol mix [49] to dye the pollen grains for increased visibility of pollen diagnostic features, and then plated for each slide. For each slide, we scanned the sample starting from the upper left corner, moving in a snake-like pattern until we encountered up to 300 grains, a common pollen identification threshold [50]. Each pollen grain encountered was identified to species level using a pollen reference library created from the anthers of all flowering species found within our sites. We only included pollen in our counts if there were more than two occurrences of it per slide [35]. When an unknown pollen grain was encountered more than two times, it was designated with a morphospecies ID [51].

### 2.5. Pollen-Based Networks

We constructed quantitative pollen-transport networks by calculating four network metrics, foraging specialization ($H_2'$), connectance, nestedness, and pollinator generality because past studies have shown that these metrics have the capacity to be influenced by local and landscape features, such as floral availability at the local scale, and amount of semi-natural habitat at the landscape scale (e.g., [17,18,33]). All metrics were calculated using the networklevel function in the bipartite package in R. Network foraging specialization ($H_2'$) describes the degree of foraging specialization between plants and pollinators while incorporating abundances into the network [52] and ranges from 0 to 1, with 0 indicating complete foraging generalization and 1 meaning complete foraging specialization. Connectance describes network complexity [53], is calculated as the proportion of observed links out of the total possible links between plants and pollinators [54], and ranges from 0 to 1 with higher values indicating higher generalization [33]. We weighted connectance, which takes 'link weights' (visitation rates) into account in order to capture the shape of the visitation distribution [55]. Nestedness describes the degree to which specialist species (those with few links) interact with generalist species (those with many links [56], and has a value of 0 to 100, where 0 represents complete randomization and 100 represents perfect nestedness [57]. We weighted nestedness by visitation rates using the NODF function [58]. Pollinator generality describes diet breadth and is calculated as the mean number of pollinators per plant species [59].

We performed all statistical analyses used in this study in the statistical program R, version 3.5.1 (R Core Team 2018). We modeled the impacts of local and landscape characteristics on bee abundance, species richness, and the four network metric response variables using linear mixed models with a Gaussian distribution in the lme4 package [60]. Because

the percentage of developed, cropland, and semi-natural habitat were highly collinear, we opted to keep only the percentage of semi-natural habitat, as in past studies [61]. Likewise, because the percent cover of leaf litter and rocky surfaces were highly collinear with the percent ground vegetation, we opted to keep the percent of live ground vegetation, as in [13,61]. Also, flower species richness was highly correlated with the mean number of flowers, thus we opted to keep the mean number of flowers in our analysis, as in [43]. After removing the five correlated variables, the remaining variables showed variance inflation scores (VIF, [62]) of approximately two or lower. Thus, the final variables we used in our models were percent ground vegetation, percent bare ground, flower abundance, amount semi-natural habitat within 2 km of each site, and habitat heterogeneity as fixed effects, as well as region (Dallas-Fort Worth vs. Austin Metroplex) as a random effect. All predictor variables were rescaled. We then tested all combinations of the fixed effect variables and conducted model selection via Akaike's information criterion (AIC) using the MuMIn package [63]. We constructed our final models by averaging all models within delta AIC < 2 of the top model [64] also using MuMIn.

### 2.6. Pollen Preference Analysis

We assessed pollen preference for the five most abundant bee species across our sites: *Halictus ligatus* Say, *Melissodes coreopsis* Robertson, *Melissodes tepaneca* Cresson, *Svastra petulca* Cresson, and *Xylocopa virginica* Linnaeus. Though *Svastra petulca* and *Melissodes tepaneca* are considered Asteraceae specialists based on the composition of pollen masses used for nest provisioning [65], nest provisions do not fully describe all floral associations by adult bees, and body pollen collected from these species has revealed that adult bees visit a variety of hosts during foraging bouts [66]. Thus, pollen-based preference studies provide a great tool for understanding foraging behavior, beyond nest provision to include nectar foraging, among both specialist and generalist bee species. Preference analyses require individual and site-level replication, which is why we focused on these five species. In order to conservatively assess pollen preference, we only included bees that had a minimum of 50 pollen grains on their bodies (as per [35]) (mean individuals collected per species: 49.1; mean number of sites per species: 11.1). Pollen preference describes a condition where a pollen species is found on a pollinator species' body significantly more than expected relative to its landscape availability [67,68] and is assessed via traditional habitat use methods [69]. We used classical compositional analyses of habitat use [24] for each species using the adehabitatHS package [70]. Specifically, to test if pollen species usage was significantly nonrandom relative to availability per site, we compared a matrix of floral species availability per site (proportion of floral cover, as measured from the vegetation surveys) with a second matrix of proportional pollen load for each plant species per bee. We evaluated the relationship between the two matrices using a randomization test (500 repetitions) where significant preference for one species over each other species was assessed [24] as in past pollen usage studies [35,67,68].

## 3. Results

Across all sites, we collected a total of 624 bees comprising 87 different species. The most common bee species collected were: *Svastra petulca* (12.8%), *Melissodes coreopsis* (9.77%), *Halictus ligatus* (7.05%), *Xylocopa virginica* (6.73%), and *Melissodes tepaneca* (5.29%). We recorded more than 35,000 inflorescences comprising 128 different plant species. Of the 624 bees, 546 carried at least 2 grains of pollen on their body. All totaled, we screened 104,000 pollen grains, with 60 species identified to species/family level, and 8 unknown morphospecies, for a total of 68 species/morphospecies used in our analysis. The most commonly encountered pollen types were *Chaerophyllum tainturieri* Hook. (13.96%), *Gaillardia pulchella* Foug. (10.04%), and *Croton monanthogynus* Michx. (9.83%). Overall, 92% of pollen found on all bees was from plant species that are native to Texas.

### 3.1. Predictors of Pollinator Community and Pollen-Based Networks

We found that foraging specialization increased with increased landscape heterogeneity ($p$ = 0.038, Table 1; Figure 1A). We also found that generality declined with increased proportion of semi-natural habitat surrounding the sites ($p$ = 0.001; Table 1; Figure 1B). Further, we found that connectance decreased with increased amounts of ground vegetation ($p$ = 0.012; Table 1; Figure 1C). We did not find any significant effect of local or landscape characteristics on nestedness ($p$ > 0.07 in all cases; Table 1). No local or landscape characteristic significantly predicted bee abundance or species richness ($p$ > 0.272; Supplementary Table S1).

**Table 1.** The final results of the averaged linear mixed models with AIC < 2 for our pollen-based plant-pollinator networks. The full models included the following predictor variables: percent ground vegetation, percent bare ground, percent flower abundance, percent semi-natural habitat within a 2 km radius, and landscape heterogeneity.

| Response | Estimate | Std. Error | Z Value | $p$ Value | $R^2$ Value |
|---|---|---|---|---|---|
| $H_2'$ | | | | | |
| Landscape heterogeneity | 0.1003 | 0.044 | 2.072 | **0.038** | – |
| $R^2$GLMM (m) | – | – | – | – | 0.262 |
| *Connectance* | | | | | |
| % Ground vegetation | −0.071 | 0.023 | 2.780 | **0.005** | – |
| % Bare ground | −0.025 | 0.024 | 0.968 | 0.333 | – |
| $R^2$GLMM (m) | – | – | – | – | 0.421 |
| *Nestedness* | | | | | |
| % Bare ground | −0.018 | 0.042 | 0.414 | 0.679 | – |
| Landscape heterogeneity | −0.029 | 0.052 | 0.544 | 0.586 | – |
| $R^2$GLMM (m) | – | – | – | – | 0.208 |
| *Generality* | | | | | |
| % Semi-natural habitat | −0.607 | 0.171 | 3.270 | **0.001** | – |
| Landscape heterogeneity | −0.157 | 0.197 | 0.203 | 0.437 | – |
| $R^2$GLMM (m) | – | – | – | – | 0.528 |

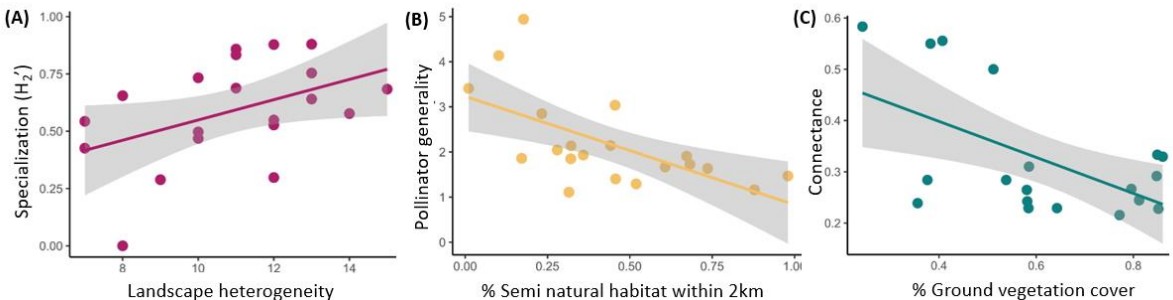

**Figure 1.** The significant relationships between the local and landscape factors and plant-pollinator network metrics: (**A**) landscape heterogeneity impacts on specialization, (**B**) % semi-natural habitat impacts on pollinator generality, and (**C**) % ground vegetation cover impacts on connectance. Grey shading represents 95% confidence intervals.

### 3.2. Pollen Preference Analysis

We found that all five focal bee species exhibited significant pollen preferences. *Halictus ligatus* ($\lambda$ = 0.209, $p$ = 0.002; Figure 2) showed the strongest preference for *Helianthus annuus* L., *Ratibida columniformis* (Nutt.) Wooton & Standl, and *Erigeron modestus* A. Gray. In contrast, *Melissodes tepaneca* ($\lambda$ = 0.456, $p$ = 0.018; Figure 2) had the strongest preference for *Convolvulus equitans* Benth., *Ratibida columniformis*, and *Gaillardia pulchella*. The congener *Melissodes coreopsis* ($\lambda$ = 0.288, $p$ = 0.002; Figure 2) had the strongest preference for *Cirsium texanum* Buckley, *Gaillardia pulchella*, and *Rudbeckia hirta*. Similarly, *Svastra petulca* ($\lambda$ = 0.377, $p$ = 0.002; Figure 2) had the strongest preference for *Gaillardia pulchella*, *Rudbeckia hirta*, and

*Helianthus annuus*. Finally, *Xylocopa virginica* (λ = 0.129, *p* = 0.002; Figure 2) had the strongest preference for *Solanum elaeagnifolium* Cav., *Convolvulus equitans*, and *Asclepias viridiflora* Raf.

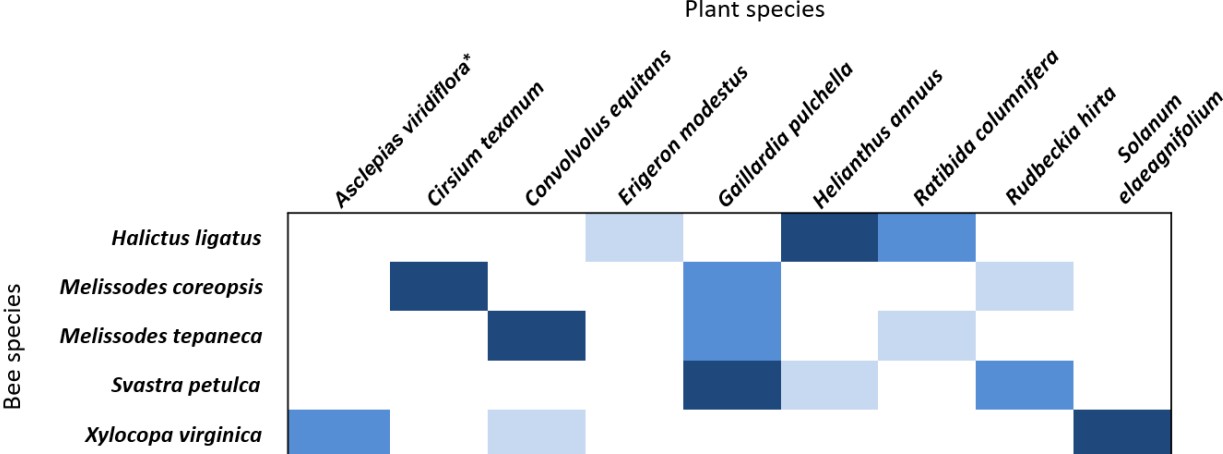

**Figure 2.** Results of the pollen preference analysis for the five most commonly occurring bee species. The color gradient represents preference with darker colors indicating higher preference and lighter colors indicating lower preference. * Denotes plants that are used solely for nectaring and not for pollen collection.

## 4. Discussion

Through the examination of pollen-based networks and pollen preference analyses, our study provides much-needed insight into bee foraging patterns and preferences across rapidly urbanizing landscapes. We found that the degree of plant-pollinator foraging specialization was significantly driven by surrounding landscape composition (2 km radius), while network connectance was driven by within-site ground vegetation. Specifically, pollinator generality was higher in sites with greater levels of surrounding semi-natural habitat cover, and pollinator specialization was higher in sites with greater landscape heterogeneity. In contrast, we found that connectance, a measure of network complexity, was lower in sites with greater ground vegetation cover. We also documented significant and distinct pollen preferences for five bee species inhabiting the same study region, highlighting the need for maintaining high flowering plant diversity to meet distinct pollinator resource needs. Our findings demonstrate the importance of landscape-level natural habitat cover and heterogeneity in shaping plant-pollinator network interactions and reveal the critical role of multiple native forb species for bee pollen collection within urbanizing landscapes.

### 4.1. Landscape-Level Habitat Composition Shapes Network Structure

First, we show that the degree of landscape heterogeneity surrounding a grassland habitat patch critically impacts the structure of plant-pollinator interaction networks. We documented greater network foraging specialization with increased landscape heterogeneity. As network foraging specialization describes niche partitioning [71,72], we show that higher habitat heterogeneity reduces the overlap of shared resources among plants and pollinators, likely facilitating inter-species coexistence due to reduced competition for both pollinators and plants. This pattern has been observed in at least one other plant-pollinator study, where greater habitat diversification across the landscape resulted in higher network foraging specialization, largely driven by segregated niches among pollinators [73]. Landscape heterogeneity can provide a variety of different nesting and feeding resources to support a functionally diverse bee community [73], which has indeed been shown to increase the number of plant species receiving visits due to niche partitioning based on pollinator functional needs [74].

In addition to greater foraging specialization, we found that pollinator generality, which describes diet breadth, declined with increased semi-natural habitat cover, indicating

that highly urbanized areas with less semi-natural habitat may drive bees to forage more generally, as seen in other human-dominated landscapes [16,75]. This outcome may be due to the fact that pollinators in human-dominated landscapes are spatially constrained to visit those plant species available within cities, even if they are not preferred, and may even experience greater metabolic costs to travel in these human-altered landscapes [76]. For example, Redhead et al., [32] found that pollinator generality was significantly higher in agriculturally dominated landscapes, where bee diet was more generalized in habitats with lower flower species richness and diversity. Similarly, past work has shown that increased landscape-level semi natural habitat cover can facilitate movement across agricultural landscapes for better foraging and nesting resources [77,78], potentially allowing bees to visit more preferred plant species.

### 4.2. Local Vegetation Drives Higher Network Connectance

The one local habitat driver of plant-pollinator network interactions was ground vegetation, where connectance was lower when live ground vegetation cover was higher. Connectance is a measure of network complexity and is associated with community stability [79,80] and past studies have found that increased floral display [34,81] and floral richness [33] may lead to greater network connectance. While we found no relationship with floral abundance, we posit that ground vegetation is likely negatively correlated with other bee nesting materials, which have also been shown to shape pollinator communities [82]. Past work has indeed shown that dead litter and reduced ground vegetation can be an important resource for stem-nesting and ground-nesting bees, respectively [83], thus greater nest proximity could increase network connectance. Because connectance is indicative of how well communities will recover from environmental disturbances [84], improving network connectance within urban grasslands could potentially buffer communities from continued land-use change. Based on our results, we posit that managing for enhanced nesting resources for bees, via the maintenance of litter or reduction of vegetation overgrowth may play an important role in shaping plant-pollinator networks. Improved pollinator nesting habitat may be particularly important in cities like Austin and Dallas-Fort Worth, where human population growth is expected to increase by over 50% between 2010 and 2050 [85], especially given that human population growth typically leads to increases in impervious ground cover [86].

### 4.3. Pollinators Show Distinct Floral Preferences for Native Flowers in Urban Landscapes

We also found that bees exhibited significant preferences for a number of native forb species, many within the Asteraceae family. Interestingly, across our five focal bee species, all of the top three preferred plant species were native to the study region. Further, out of all the pollen found on bees, only 5% of the plant species were non-native to Texas, showing that bees in our system primarily forage on native plants, as seen in more rural grasslands within our study region [35]. This pattern has also been documented in agricultural hedgerows on the US west coast [87], and remnant and restored heathlands in Europe [88], among other systems. While the five bee species in our study differed in their preferences, most preferred to forage on Asteraceae plants, with a number of bee species focusing on *Gaillardia pulchella*, and *Rudbekia hirta*. One caveat to our study is that though *Svastra petulca* and *Melissodes tepaneca* are Asteraceae specialists, the other three bee species used in our pollen preference study are generalist foragers, and their pollen preference might vary based on resources and across systems with different flowering species [89]. However, our findings are similar to another pollen preference study conducted within more rural grasslands in the same bioregion [35]. Further, several other studies conducted throughout South America, Europe, and South Africa have also documented Asteraceae plants as important resources for bees across ecosystems [90–93]. Asteraceae flowers have compound inflorescences that yield high amounts of pollen and nectar resources for bees and often have a colorful and large floral display [93], which might make them attractive to many different bee species. For example, *G. pulchella* is prevalent across Central Texas and

has been found in pollen stores in nests of both specialist and generalist bees indicating that this is an important pollen source for a diverse bee assemblage [94]. Finally, though *Xylocopa virginica* bees were both collected from and shown to prefer *Asclepias viridiflora* in our study, bees do not actively collect pollen from milkweed [95]. Instead, milkweed is a rich nectar resource for bees [95] so it is likely that *Xylocopa virginica* picked up milkweed pollen while nectaring. Thus, sites containing a variety of Asteraceae species as well as plants known to offer high nectar rewards will likely support a more diverse bee community [96].

### 4.4. Conclusions and Conservation Applications

Overall, our study reveals the importance of both local and landscape habitat management for the structure of plant-pollinator interaction networks. We also highlight the critical role of native forbs for native bee pollen collection within urbanizing grassland ecoregions. Based on our findings, we suggest that land managers conserve and promote semi-natural habitat and native plant abundance within urban areas, not only because native plants make up most native bee pollen collections, but also because our network results indicate that pollinators become more specialized as natural habitat becomes more available. These recommendations could also benefit the plant community given that higher pollinator foraging specialization [97], and greater natural cover [98,99] could increase conspecific pollen deposition on stigmas. Our results and others indicate that maintaining native floral resources and natural habitat patches can support native pollinators and their foraging interactions across rapidly urbanizing landscapes.

**Supplementary Materials:** The following supporting information can be downloaded at: https://www.mdpi.com/article/10.3390/land12020362/s1, Table S1: Results of local and landscape drivers on bee abundance and species richness.

**Author Contributions:** Conceptualization, G.L.P., K.M.B. and S.J.; methodology, K.M.B., L.Q.D., D.O., J.L.N. and S.J.; software, G.L.P.; validation, G.L.P., K.M.B. and S.J.; formal analysis, G.L.P. and S.J.; investigation, G.L.P., K.M.B. and S.J.; resources, G.L.P., K.M.B., L.Q.D., D.O., J.L.N. and S.J.; data curation, G.L.P., K.M.B., L.Q.D., D.O., J.L.N. and S.J.; writing—original draft preparation, G.L.P. Pardee; writing—review and editing, G.L.P., K.M.B., J.L.N., L.Q.D., D.O., E.J.B., B.J.B., T.H.G., J.A.M., D.T. and S.J.; visualization, G.L.P.; supervision, G.L.P., K.M.B. and S.J.; project administration, G.L.P., K.M.B. and S.J.; funding acquisition, K.M.B., E.J.B., B.J.B., T.H.G., J.A.M., D.T. and S.J. All authors have read and agreed to the published version of the manuscript.

**Funding:** This research was funded by Texas Parks and Wildlife Department, Texas Ecolabs, the Native Plant Society of Texas Ann Miller Gonzales Graduate Research Grant, and the Graduate Program in Ecology, Evolution, and Behavior at the University of Texas at Austin and the National Science Foundation. Additionally, the research reported here was funded in whole or in part by DEVCOM ARL, ARO through a Multidisciplinary University Research Initiative Grant (#W911NF1910231). The research, interpretations, and perspectives reported here are those of the authors and should not be attributed to the Army or the Department of Defense.

**Data Availability Statement:** Data available upon request from corresponding author.

**Acknowledgments:** We would like to thank individual landowners, Texas State Parks, and the Lower Colorado River Authority for allowing access to sites. We thank R. Ruppel, C. Glinka, A. Ritchie, K. Merrill, B. French, N. Vojnovich, M. Rolbiecki, T. Ortega, L. Stevens, and N. Fogel for assistance with the field collection of insects and vegetation surveys, and assistance with insect preservation and curation.

**Conflicts of Interest:** The authors declare no conflict of interest.

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
