# Peer review of "Local and Landscape Factors Influence Plant-Pollinator Networks and Bee Foraging Behavior across an Urban Corridor"

_land, doi:10.3390/land12020362_

Round 1

Reviewer 1 Report

Dear authors,

I enjoyed reading your study, an impressive piece of work, which I think will make an important contribution to the plant-pollinator network-literature. However, while reading the manuscript I was however left wondering about two things which I feel should be addressed:

(1) Please provide a bit more detail in your sampling description, to inform the reader how you avoided cross-contaminaiton between bee specimens.

(2) Either (a) split the seminatural land cover category into two parts: forest vs. grassland/shrubland; or (b) provide information on why this split was not nessecary, i.e. that forest cover is so low that you find herbaceous floral resources also in your forest patches.

Specific comments:

L39-41: While I would agree that ecologists have only recently begon studying how species interactions and interaction networks change along environmental gradients (incl. in human modified landscapes), this statement is a bit strong. see e.g. Pellissier, L., Albouy, C., Bascompte, J., Farwig, N., Graham, C., Loreau, M., ... & Gravel, D. (2018). Comparing species interaction networks along environmental gradients. Biological Reviews, 93(2), 785-800. and studies that cite this.

L58-59: I don't think Olesen et al. 2007 is the correct reference for this statement.

L96-97: A hypothesized increase in both network level specialization AND pollinator generality seems counter-intuitive. Add a bit more context to these hypotheses/predictions.

L99-100: could an increased connectivitity with floral abundance not be indicative of a neutral network assembly? i.e. the more abundant a resource is the more likely it is to attract pollinators? Vázquez, D. P., Blüthgen, N., Cagnolo, L., & Chacoff, N. P. (2009). Uniting pattern and process in plant–animal mutualistic networks: a review. Annals of botany, 103(9), 1445-1457. and Sydenham, M. A., Venter, Z. S., Moe, S. R., Eldegard, K., Kuhlmann, M., Reitan, T., ... & Rusch, G. M. (2022). Neutral processes related to regional bee commonness and dispersal distances are important predictors of plant–pollinator networks along gradients of climate and landscape conditions. Ecography, 2022(12), e06379.

L138: please add a bit more information about the NLCD dataset - what is the resolution and minimum mapping unit?

L155-180: From the sampling description I could not work out how you avoided cross contamination. Please add information on how you avoided pollen being moved across specimens.

L200: Nestedness is not mentioned with the other hypotheses/predictions in the intro

L147: The semi natural land category seems too inclusive to me. Please split so that forest becomes as class in itself. Forest cover and grassland likely affect bee diversity in opposite ways e.g. Winfree, R., Griswold, T., & Kremen, C. (2007). Effect of human disturbance on bee communities in a forested ecosystem. Conservation biology, 21(1), 213-223.

L220: in the tables you refer to AICc ant not AIC. please check for consistency.

L222: Please consider adding an R2 value (since you are already using MuMIn) to your models.

L290: Here and elsewhere avoid the use 'predicted' since you are not predicting onto withheld/new data (which is fine!).

Author Response

Our responses are in bold text

Dear authors,

I enjoyed reading your study, an impressive piece of work, which I think will make an important contribution to the plant-pollinator network-literature. However, while reading the manuscript I was however left wondering about two things which I feel should be addressed:

(1) Please provide a bit more detail in your sampling description, to inform the reader how you avoided cross-contamination between bee specimens.

We added a sentence that described how we avoided cross contamination, (lines 161-163): The sentence now reads “To reduce cross-contamination, each bee was placed into a separate vial upon collection, and the observer shook out the hand net to remove any pollen”

(2) Either (a) split the seminatural land cover category into two parts: forest vs. grassland/shrubland; or (b) provide information on why this split was not nessecary, i.e. that forest cover is so low that you find herbaceous floral resources also in your forest patches.

 This comment is addressed further down in the “specific comments” section

Specific comments:

L39-41: While I would agree that ecologists have only recently begon studying how species interactions and interaction networks change along environmental gradients (incl. in human modified landscapes), this statement is a bit strong. see e.g. Pellissier, L., Albouy, C., Bascompte, J., Farwig, N., Graham, C., Loreau, M., ... & Gravel, D. (2018). Comparing species interaction networks along environmental gradients. Biological Reviews, 93(2), 785-800. and studies that cite this.

Changed sentence to: “Despite this fact, researchers have only recently begun to examine how changes in habitat composition affect mutualistic relationships across human-modified landscapes” and added the recommended citation.

L58-59: I don't think Olesen et al. 2007 is the correct reference for this statement.

           Changed citation to Spiesman, B.J.; Inouye, B.D. Habitat Loss Alters the Architecture of Plant-Pollinator Interaction Networks. Ecology 2013, 94, 2688–2696.

L96-97: A hypothesized increase in both network level specialization AND pollinator generality seems counter-intuitive. Add a bit more context to these hypotheses/predictions.

We changed sentence to: “We hypothesized that network specialization would increase and pollinator generality would decrease with increased amount of surrounding semi natural habitat, indicative of more selective foraging behavior, as seen in past studies.”

L99-100: could an increased connectivitity with floral abundance not be indicative of a neutral network assembly? i.e. the more abundant a resource is the more likely it is to attract pollinators? Vázquez, D. P., Blüthgen, N., Cagnolo, L., & Chacoff, N. P. (2009). Uniting pattern and process in plant–animal mutualistic networks: a review. Annals of botany, 103(9), 1445-1457. and Sydenham, M. A., Venter, Z. S., Moe, S. R., Eldegard, K., Kuhlmann, M., Reitan, T., ... & Rusch, G. M. (2022). Neutral processes related to regional bee commonness and dispersal distances are important predictors of plant–pollinator networks along gradients of climate and landscape conditions. Ecography, 2022(12), e06379.

This is a good point. We changed this sentence to include vegetation cover as well to show that a combination of high floral abundance and vegetation cover could increase connectance through greater habitat complexity.

L138: please add a bit more information about the NLCD dataset - what is the resolution and minimum mapping unit?

Added: 30 m resolution and minimum mapping areas of five pixels

L155-180: From the sampling description I could not work out how you avoided cross contamination. Please add information on how you avoided pollen being moved across specimens.

We added a sentence that described how we avoided cross contamination, lines 161-163: The sentence now reads “To reduce cross-contamination, each bee was placed into a separate vial upon collection, and the observer shook out the hand net to remove any pollen”

L200: Nestedness is not mentioned with the other hypotheses/predictions in the intro

Added a prediction for nestedness

L147: The semi natural land category seems too inclusive to me. Please split so that forest becomes as class in itself. Forest cover and grassland likely affect bee diversity in opposite ways e.g. Winfree, R., Griswold, T., & Kremen, C. (2007). Effect of human disturbance on bee communities in a forested ecosystem. Conservation biology, 21(1), 213-223.

 We combined forest cover and grassland into one category in order to keep our landscape metric consistent with a previously published paper that was conducted on bee communities at the same sites: Ballare, K.M.; Neff, J.L.; Ruppel, R.; Jha, S. Multi-Scalar Drivers of Biodiversity: Local Management Mediates Wild Bee Community Response to Regional Urbanization. Ecological Applications 2019, 29, e01869, doi:10.1002/eap.1869. The Ballare (2019) study is the only other urban community-level study conducted in the region, so it was essential to examine if the landscape predictors were the same. Grassland and forest are part of a fine-scale interdigitated mosaic of habitat in central Texas, and both categories were highly colinear in both bee and floral diversity which why they were combined into one metric in the Ballare 2019 study.

L220: in the tables you refer to AICc ant not AIC. please check for consistency.

Changed from AICc to AIC

L222: Please consider adding an R2 value (since you are already using MuMIn) to your models.

R2 values were added in the table

L290: Here and elsewhere avoid the use 'predicted' since you are not predicting onto withheld/new data (which is fine!).

Changed “predicted” to “driven by”

Reviewer 2 Report

General comments: The work is interesting and has valuable impacts on vegetation management and bee conservation. However, revision is needed to improve the manuscript. References in the text should be given as per the journal's style (using the number system). Reference list will be as per the journal's style.

Specific comments: Given in the attached PDF file and also in below:

Line 31: Keywords--- arrange alphabetically.

Line 38-39: References in the text will be in a number system.

Line 53-54: add more references.

Line 76-81. Pollen host selection for a bee species depends on complex parameters, including flower availability and pollen availability (Layek et al. 2020; Palynology 44(1), 114-126. doi: 10.1080/01916122.2018.1533898). Consider this in this section.

Line 91-92: add more references. Winfree et al. 2007; Ecology letters 10, 1105-1113. doi: 10.1111/j.1461-0248.2007.01110.x

Line 110-111: rewrite.

Line 68-75: Is pollen grains are processed to increase the visibility of diagnostic characters? 

Line 179-180: add reference for pollen type system of classification. Joosten and Klerk 2002 Review of Palaeobotany and Palynology 122, 29-45. doi: 10.1016/S0034-666

Line 225-226: add author citations for scientific names in first time within the text.

Line 250: pollen species--- pollen types

Line 255-256: table 1, fig. 1--- maintain journal's style like Table 1, Figure 1

Line 336-337: reconsider it because abundance and diversity of floral visitors largely depends on surrounding vegetation. Senapathi et al. 2016 Functional Ecology 31(1), 26-37. doi: 10.1111/1365-2435.12809; Gilpin et al. 2022 Agriculture 12, 1246. doi: 10.3390/agriculture12081246

References list---- maintain proper style as applicable for the journal.

Author Response

Our responses to each comment are in bold text

General comments: The work is interesting and has valuable impacts on vegetation management and bee conservation. However, revision is needed to improve the manuscript. References in the text should be given as per the journal's style (using the number system). Reference list will be as per the journal's style.

Thanks for taking the time to review our manuscript! We have changed the citation style in accordance with Land citation style

Specific comments: Given in the attached PDF file and also in below:

Line 31: Keywords--- arrange alphabetically.

Arranged alphabetically

Line 38-39: References in the text will be in a number system.

Changed to number system based on Land citation style

Line 53-54: add more references.

Added this reference: Klein, A.M.; Vaissière, B.E.; Cane, J.H.; Steffan-Dewenter, I.; Cunningham, S.A.; Kremen, C.; Tscharntke, T. Importance of Pollinators in Changing Landscapes for World Crops. Proceedings of the Royal Society B: Biological Sciences 2007, 274, 303–313.

Line 76-81. Pollen host selection for a bee species depends on complex parameters, including flower availability and pollen availability (Layek et al. 2020; Palynology 44(1), 114-126. doi: 10.1080/01916122.2018.1533898). Consider this in this section.

Thank you for bringing up this point. We have added a caveat to our discussion to state that foraging behavior of the 5 bees used in our study might change depending on the factors you mentioned. The sentence reads (lines 336-339): “One caveat to our study is that though Svastra petulca and Melissodes tepaneca are Asteraceae specialists, the other three bee species used in our pollen preference study are generalist foragers, and their pollen preference might vary based available resources and across systems with different flowering species [84]”

Line 91-92: add more references. Winfree et al. 2007; Ecology letters 10, 1105-1113. doi: 10.1111/j.1461-0248.2007.01110.x

Added the suggested reference

Line 100-101: rewrite.

Rewrote to make more clear, text now reads: “while network connectance would increase with increased floral abundance, as high floral abundance is often indicative of greater habitat complexity and food resources [32,33].”

Line 68-75: Is pollen grains are processed to increase the visibility of diagnostic characters? 

Yes, we stained the pollen with fuchsin to increase visibility of pollen diagnostic features. We changed this in the sentence, so it now reads “was added to 60 microliters of fuchsin + gylcerol mix [45] to dye the pollen grains for increased visibility of pollen diagnostic features”

Line 179-180: add reference for pollen type system of classification. Joosten and Klerk 2002 Review of Palaeobotany and Palynology 122, 29-45. doi: 10.1016/S0034-666

Citation inserted

Line 225-226: add author citations for scientific names in first time within the text.

Author citations added for all scientific species listed in the methods and results

Line 250: pollen species--- pollen types

Changed to “types”

Line 255-256: table 1, fig. 1--- maintain journal's style like Table 1, Figure 1

Changed all to match the journal’s style

Line 336-337: reconsider it because abundance and diversity of floral visitors largely depends on surrounding vegetation. Senapathi et al. 2016 Functional Ecology 31(1), 26-37. doi: 10.1111/1365-2435.12809; Gilpin et al. 2022 Agriculture 12, 1246. doi: 10.3390/agriculture12081246

We have added the Senapathi citation.

References list---- maintain proper style as applicable for the journal.

Changed to match journal’s citation style

Reviewer 3 Report

Pollen analysis performed on an uncertain number of bee individuals captured on particular flowering species likely contained such pollen species in abundance. Therefore, I doubt that the methodology for estimating preference (collected vs available resources) has been correctly addressed. Furthermore, the five selected bee species (selected because they were found at all study sites and had pollen on their bodies) are not necessarily pollen specialists nor do they prefer such plants, but only due to sampling bias.

Network analysis is partly adequate, but is based on incorrect and biased methodology. For example, the network parameters specialization and generality. The authors suggested that generality described diet breadths of the bees studied, but based on load sampled captured on particular pollen plants??

Author Response

Our responses to each comment are in bold text

Pollen analysis performed on an uncertain number of bee individuals captured on particular flowering species likely contained such pollen species in abundance. Therefore, I doubt that the methodology for estimating preference (collected vs available resources) has been correctly addressed. Furthermore, the five selected bee species (selected because they were found at all study sites and had pollen on their bodies) are not necessarily pollen specialists nor do they prefer such plants, but only due to sampling bias.

The reviewer brings up a valid point, though two of the five species used in our pollen preference study were Asteraceae specialists. The remaining three species are not pollen specialists and forage generally on plants, but previous studies have shown that even generalist pollinators can show preference for specific plant species:

 Ritchie, A.D.; Ruppel, R.; Jha, S. Generalist Behavior Describes Pollen Foraging for Perceived Oligolectic and Polylectic Bees. Environ Entomol 2016, 45, 909–919, doi:10.1093/ee/nvw032.

 It is also true that the generalist bees at our sites may show preference for plant species based on the resources that are available and that their preferences could change in other systems with different flowering communities. We have added a caveat about this in the discussion on lines 336-338: “One caveat to our study is that though Svastra petulca and Melissodes tepaneca are Asteraceae specialists, the other three bee species used in our pollen preference study are generalist foragers, and their pollen preference might vary based available resources and across systems with different flowering species [84].”  We also report the number of individuals collected in the methods section (lines 234-243).

Network analysis is partly adequate, but is based on incorrect and biased methodology. For example, the network parameters specialization and generality. The authors suggested that generality described diet breadths of the bees studied, but based on load sampled captured on particular pollen plants??

We thank the reviewer for raising this concern and now include additional justification for our approach. Yes, diet breadth was calculated based on the pollen load sampled from individuals collected while netting. Though we agree that some of the bees we collected likely carried higher pollen abundance of the plant it was collected from, we were still able infer diet breadth based on other pollen grain species found on the bodies. This method of measuring diet breadth through pollen found on bodies has been used extensively in past studies, which we’ve cited throughout the manuscript. Some examples of these citations are:

Vaudo, A.D.; Tooker, J.F.; Patch, H.M.; Biddinger, D.J.; Coccia, M.; Crone, M.K.; Fiely, M.; Francis, J.S.; Hines, H.M.; Hodges, M.; et al. Pollen Protein: Lipid Macronutrient Ratios May Guide Broad Patterns of Bee Species Floral Preferences. Insects 2020, 11, 132, doi:10.3390/insects11020132.

Barker, D.A.; Arceo-Gomez, G. Pollen Transport Networks Reveal Highly Diverse and Temporally Stable Plant-Pollinator Interactions in an Appalachian Floral Community. AoB Plants 2021, 13, plab062, doi:10.1093/aobpla/plab062.

Ritchie, A.D.; Ruppel, R.; Jha, S. Generalist Behavior Describes Pollen Foraging for Perceived Oligolectic and Polylectic Bees. Environ Entomol 2016, 45, 909–919, doi:10.1093/ee/nvw032

Jha, S.; Stefanovich, L.; Kremen, C. Bumble Bee Pollen Use and Preference across Spatial Scales in Human-Altered Landscapes. Ecol Entomol 2013, 38, 570–579, doi:10.1111/een.12056.

Reviewer 4 Report

This is a very interesting manuscript which identifies what is a global problem as an increasing human population becomes more and more concentrated in cities, requiring rapid expansion of the area of cities, both for residential and office/industrial purposes . While the expansion of cities is a global problem the areas in which those cities are expanding will differ both in their environmental conditions and the biota which would naturally occur in them. My main concern which I will raise amongst the points I make below is that what happens in Texas, which is important in consideration of conservation and maintenance of agriculture close to cities in Texas, maybe different in other parts of the world. Texas might provide an indication of the sorts of things that need to be looked at in other parts of the world but some of the detailed findings from Texas may not be replicated elsewhere. 

Line 16

‘human- mediated bee declines’

arel the declines in the number of bee species, or in the total number of individual bees or both? 

Line 24 

‘more than 300 kilometres of urban habitat’ -is 350 square kilometres what was intended?

Line 26 

Urban grasslands 

These are important habitats in many urban areas, but across the world there are many other habitats represented in urban and near urban areas in particular where urbanisation is expanding directly into heath, woodland or forest [including tropical rainforest] without an intervening agricultural phase, either of cropping land or grazed semi natural grassland.

 These habitats may support different pollinator faunas- and conservation of  these other communities should play an important role in regional biodiversity conservation strategies.

 Line 29 

 Asteraceae family- inclusion of family is redundant, the Asteraceae is a family. More relevantly Asteraceae may be a feature of the urban fringe in Texas, but would not be as important in many other locations. 

Line 35

urban, suburban, and peri- urban habitats

Perhaps explanation of the differences between these three terms would be useful. 

Long established suburbs in northern temperate cities may be characterised by gardens which include important resources for pollinators. Mew suburbs globally may be much more densely populated, and high rise, and individual gardens associated with dwellings are rare - in this case the differentiation between urban and suburban areas maybe more one of  function than form. 

Peri-urban could take a variety of meanings and forms and include still functioning agricultural land destined for future urban development. Many major towns and cities in northern Europe have long been surrounded by agricultural land and expansion of cities in the 20th and 21st centuries is at the expense of agricultural land. with a loss of local human food production and the need for development of new supply chains with consequences for the structure of agriculture globally. However, some of these old cities retain areas of natural seminatural communities within their boundaries, for example Hampstead Heath and Epping Forest in London. More recently established europeanise cities elsewhere, for example the state capitals of Australian states, have not only examples of near intact natural bush enclaves within urban areas but the expanding outer margin of the cities more frequently is into surrounding bush rather than to agricultural land. These different geographies are likely to affect the composition of the surviving .native fauna, including pollinating organisms.

 Lines 53 the 54 

The data on how plants are pollinated are still sparse and the inferences that can be drawn from the data might depend on the context in which the question is asked.

 In terms of how many species of direct economic value to humans then in temperate forests trees utilised for timber production. both gymnosperms and angiosperms, are wind pollinated whereas in tropical forests many trees are animal pollinated, not just by insects but for some species by birds or bats.

In forests and pastures what is economically important to humans is only a small proportion of the total flora, but they are species with very large populations whereas the vast majority of species have relatively small populations. The plant- pollinator relationship vary from highly specialised 1-1 relationships to the very generalised, almost anything goes. 

 Ollerton et al 2011 is an important paper but it is one which if  gone through with a fine tooth comb reveals how little we actually know compared to what we assume. 

 Line 66 

‘human- dominated landscapes’

Are there any landscapes today for which it could be argued that they are not human- dominated’? Human induced climatic change, for example, is universal in its impacts including on pollinator relationships. 

 Line 89

350 km -is this linear or area 

Line 103  

Asteraceae may be particularly important in the authors’ study region, but while the family has a very wide distribution and a very large number of species, there are many circumstances where important communities in the urban fringe, such as heathland, where the number of indigenous species of Asteraceae is small and other families with very different floral structures predominate.

Line 107

350 kilometres

Will the majority of international readers have any idea what is meant by the I-35 Metroplex.

 Line 100 and 113

-this implies that metro and Metroplex are synonymous.

 Line 115 to 116 

 The absence of disturbance,  for example, no recent fire, may of itself when viewed over a longer time period be recognised as a disturbance. Alteration to fire regimes is considered in many parts of the world to be a threatening process to both ecosystems and species.

Line 119 120 

 Bee foraging ranges vary considerably,  with some specialist rainforest bees having much longer ranges as was shown many years ago by Dan Janzen. In systems where bees play a lesser role the foraging range of pollinators from different taxa maybe much less ,or  much larger. Megachiroptera [fruit bats], important pollinators of some rainforest trees and some eucalyptus, have a foraging range of up to 50 kilometres.   (Fruit bats probably have a more imortant role as seed dispersers, but their contribution to pollination is not negligible

Lines 156 -158 

Apis melifera is not native -but are there feral populations in the study area. Feral honey bees importantly compete with native fauna -with birds and mammals for habitats in tree hollows, with native pollinators for rewards, and with hive bees for resources. Additionally, feral bees may act as sources of pathogens and parasites which affect managed bees - such as being a vector for Varroa mite spread.

In the current manuscript I think there needs to be an expanded justification for ignoring honey bees in the study. If there are large numbers of honey bees, either Feral or from managed hives, the numbers of native pollinators may be substantially reduced so that both the number of species and the number of individuals recorded may have been far lower then in the pre honey bee era,and some native species may have been completely lost, at least at the local scale and recovery may not be possible.

Line 212

Flower species richness. This means the number of flowering plant species, but are there patterns in diversity of flowers in features such as the number of particular colours, or different flower morphological types ?

 line 276 

‘contained at least two grains of pollen’

 but the pollen grains are not contained within, rather to be of value in the plant reproductive process they are carried externally.

Do the data demonstrate preference or do they reflect the relative abundance of particular species/morphological types of flower which may vary overtime and between years?

 References 

The references listed are in themselves an extremely valuable resource for other workers. The citations of journal references seem to be in an appropriate and consistent style, but for reports and book chapters there is variation. I would presume that the journal has instructions it applies to how such works are to be cited. I won't identify all but some examples are:

 line 448 

who were the editor(s) and what is the place of publication

 line 467

 Is this a hard copy production or is it a website 

 line 492

The editors are listed but what are the publication details

Line 528 

What are the publication details

Author Response

Our responses to each comment are in bold text

This is a very interesting manuscript which identifies what is a global problem as an increasing human population becomes more and more concentrated in cities, requiring rapid expansion of the area of cities, both for residential and office/industrial purposes . While the expansion of cities is a global problem the areas in which those cities are expanding will differ both in their environmental conditions and the biota which would naturally occur in them. My main concern which I will raise amongst the points I make below is that what happens in Texas, which is important in consideration of conservation and maintenance of agriculture close to cities in Texas, maybe different in other parts of the world. Texas might provide an indication of the sorts of things that need to be looked at in other parts of the world but some of the detailed findings from Texas may not be replicated elsewhere. 

Thank you for your valuable comments. We have expanded some of our findings to show that our results are not only applicable to Texas. For example, our findings that Asteraceae pollen is a valuable resource for pollinators has been found in other ecosystems beyond grasslands and in studies conducted in Europe, South America, and Africa. We have added a sentence in the discussion (lines 339-343) to put our findings into broader contexts. Further, our findings that native plants are preferred by bees in our grasslands has been shown in other study systems as well, which we highlight in lines 331-333.

Line 16

‘human- mediated bee declines’ are the declines in the number of bee species, or in the total number of individual bees or both? 

 Edited to include both abundance and richness

Line 24 

‘more than 300 kilometres of urban habitat’ -is 350 square kilometres what was intended?

The wording is correct here

Line 26 

Urban grasslands 

These are important habitats in many urban areas, but across the world there are many other habitats represented in urban and near urban areas in particular where urbanisation is expanding directly into heath, woodland or forest [including tropical rainforest] without an intervening agricultural phase, either of cropping land or grazed semi natural grassland. These habitats may support different pollinator faunas- and conservation of  these other communities should play an important role in regional biodiversity conservation strategies.

We agree with this reviewer that other types of natural habitat (heathland, forest, woodland, etc) within urban areas are important areas for bees. However, for the purpose of this study we exclusively sampled in grasslands, which is why we used this wording in the abstract and throughout our study.

 Line 29 

 Asteraceae family- inclusion of family is redundant, the Asteraceae is a family. More relevantly Asteraceae may be a feature of the urban fringe in Texas, but would not be as important in many other locations. 

Removed “family”

Line 35

urban, suburban, and peri- urban habitats

Perhaps explanation of the differences between these three terms would be useful. 

Long established suburbs in northern temperate cities may be characterised by gardens which include important resources for pollinators. Mew suburbs globally may be much more densely populated, and high rise, and individual gardens associated with dwellings are rare - in this case the differentiation between urban and suburban areas maybe more one of  function than form. 

Peri-urban could take a variety of meanings and forms and include still functioning agricultural land destined for future urban development. Many major towns and cities in northern Europe have long been surrounded by agricultural land and expansion of cities in the 20th and 21st centuries is at the expense of agricultural land. with a loss of local human food production and the need for development of new supply chains with consequences for the structure of agriculture globally. However, some of these old cities retain areas of natural seminatural communities within their boundaries, for example Hampstead Heath and Epping Forest in London. More recently established europeanise cities elsewhere, for example the state capitals of Australian states, have not only examples of near intact natural bush enclaves within urban areas but the expanding outer margin of the cities more frequently is into surrounding bush rather than to agricultural land. These different geographies are likely to affect the composition of the surviving .native fauna, including pollinating organisms.

Thank you for bringing this to our attention as a potential cause for confusion. Upon reflection, we felt this phrase unnecessary to understanding our study and removed it from the introduction. The sentence now reads: “The urban to rural interface has long been shown to drive changes in animal community composition via shifts in food and nesting resources, alteration of biotic and abiotic factors, and the impediment of an organism’s ability to disperse into hospitable habitat [1,2].”

 Lines 53 the 54 

The data on how plants are pollinated are still sparse and the inferences that can be drawn from the data might depend on the context in which the question is asked.

 In terms of how many species of direct economic value to humans then in temperate forests trees utilised for timber production. both gymnosperms and angiosperms, are wind pollinated whereas in tropical forests many trees are animal pollinated, not just by insects but for some species by birds or bats.

In forests and pastures what is economically important to humans is only a small proportion of the total flora, but they are species with very large populations whereas the vast majority of species have relatively small populations. The plant- pollinator relationship vary from highly specialised 1-1 relationships to the very generalised, almost anything goes. 

 Ollerton et al 2011 is an important paper but it is one which if  gone through with a fine tooth comb reveals how little we actually know compared to what we assume. 

We agree that the data are still sparse on how many plants are pollinated, but this reference was used to demonstrate the importance of plant-pollinator relationships for pollination and plant reproduction. We included an additional reference as well here, based on the recommendation of reviewer 1: Klein, A.M.; Vaissière, B.E.; Cane, J.H.; Steffan-Dewenter, I.; Cunningham, S.A.; Kremen, C.; Tscharntke, T. Importance of Pollinators in Changing Landscapes for World Crops. Proceedings of the Royal Society B: Biological Sciences 2007, 274, 303–313.

 Line 66 

‘human- dominated landscapes’

Are there any landscapes today for which it could be argued that they are not human- dominated’? Human induced climatic change, for example, is universal in its impacts including on pollinator relationships. 

We used “human-dominated” here to describe urbanizing environments that have high population growth. Both references cited here describe the effects of urbanization on plant-pollinator interactions.

 Line 89

350 km -is this linear or area 

Linear. We used corridor here to describe this area, but also included linear to make it more clear. It now reads “a ~350 km rapidly urbanizing linear corridor in the southern US”

Line 103  

Asteraceae may be particularly important in the authors’ study region, but while the family has a very wide distribution and a very large number of species, there are many circumstances where important communities in the urban fringe, such as heathland, where the number of indigenous species of Asteraceae is small and other families with very different floral structures predominate.

We specified that we are basing this prediction off of other studies that were conducted in TX to show that Asteraceae plants might be more influential in this region compared to others The sentence now reads  “Asteraceae, as seen in other pollen preference studies conducted in Texas, US [34]

Line 107

350 kilometres

Will the majority of international readers have any idea what is meant by the I-35 Metroplex.

Changed km to kilometres. We added “highway” after I-35 so international readers will know this is a highway that spans across 350 km and encompasses five of the 20 fastest growing cities in the US.

 Line 100 and 113

-this implies that metro and Metroplex are synonymous.

Changed to metroplexes for consistency

 Line 115 to 116 

 The absence of disturbance,  for example, no recent fire, may of itself when viewed over a longer time period be recognised as a disturbance. Alteration to fire regimes is considered in many parts of the world to be a threatening process to both ecosystems and species.

This is true, however the purpose of this statement was to describe how the habitats we sampled were similar in terms of their management efforts to reduce sampling biases.

Line 119 120 

 Bee foraging ranges vary considerably,  with some specialist rainforest bees having much longer ranges as was shown many years ago by Dan Janzen. In systems where bees play a lesser role the foraging range of pollinators from different taxa maybe much less ,or  much larger. Megachiroptera [fruit bats], important pollinators of some rainforest trees and some eucalyptus, have a foraging range of up to 50 kilometres.   (Fruit bats probably have a more important role as seed dispersers, but their contribution to pollination is not negligible

The paper we cited (Greenleaf, 2007) describes how bee foraging range is correlated with body size. Though the reviewer brings up a good point that there are some exceptions to this rule, the majority of bee species in our system are small enough and generalist foragers that they likely won’t travel more than 2km from their nest, based on the Greenleaf paper.

Lines 156 -158 

Apis melifera is not native -but are there feral populations in the study area. Feral honey bees importantly compete with native fauna -with birds and mammals for habitats in tree hollows, with native pollinators for rewards, and with hive bees for resources. Additionally, feral bees may act as sources of pathogens and parasites which affect managed bees - such as being a vector for Varroa mite spread.

In the current manuscript I think there needs to be an expanded justification for ignoring honey bees in the study. If there are large numbers of honey bees, either Feral or from managed hives, the numbers of native pollinators may be substantially reduced so that both the number of species and the number of individuals recorded may have been far lower then in the pre honey bee era,and some native species may have been completely lost, at least at the local scale and recovery may not be possible.

 Honey bees are regularly excluded from consideration in community studies, like the one we cited, since their numbers can vary widely depending on the placement of managed hives. Feral colonies typically occur at low densities in our study area and there were no managed hives within our sites.  Though we acknowledge that there may be foraging interactions between honey bees and native bees, this was beyond the scope of our paper.

Line 212

Flower species richness. This means the number of flowering plant species, but are there patterns in diversity of flowers in features such as the number of particular colours, or different flower morphological types?

Yes, other patterns in flower diversity such as differences in plant traits might affect plant-pollinator interactions but examining plant traits was outside the scope of our study. We used flower abundance as a local habitat feature based on previous studies that have examined local vs landscape effects on plant-pollinator networks, as referenced throughout our manuscript.

 line 276 

‘contained at least two grains of pollen’

 but the pollen grains are not contained within, rather to be of value in the plant reproductive process they are carried externally.

We changed the wording, specifically from “contained” to “carried” to make this sentence clearer. It now reads “546 carried at least 2 grains of pollen on their body”

Do the data demonstrate preference or do they reflect the relative abundance of particular species/morphological types of flower which may vary overtime and between years?

The data was used to demonstrate preference, though we recognize that preference might change from year to year, depending on floral abundance. We have added a caveat about this in the discussion on lines 336-338: “One caveat to our study is that though Svastra petulca and Melissodes tepaneca are Asteraceae specialists, the other three bee species used in our pollen preference study are generalist foragers, and their pollen preference might vary based available resources and across systems with different flowering species [84].”

 References 

The references listed are in themselves an extremely valuable resource for other workers. The citations of journal references seem to be in an appropriate and consistent style, but for reports and book chapters there is variation. I would presume that the journal has instructions it applies to how such works are to be cited. I won't identify all but some examples are:

Thank you for bringing these discrepancies to our attention. We have changed the references you mentioned below and went through the reference section to make sure all the citations are correct.

 line 448 

who were the editor(s) and what is the place of publication

Changed to include editors

 line 467

 Is this a hard copy production or is it a website 

Website, but this is how it was cited in other publications

 line 492

The editors are listed but what are the publication details

Changed to include publication details

Line 528 

What are the publication details

This is how it was cited in previously published articles.

Reviewer 5 Report

Research on ecological corridors for urban fauna and semi-natural habitats outside cities, including in agricultural landscapes, is extremely valuable. They have been carried out in Europe for many years. Using these corridors, both pollinators and natural enemies of pests can move - disperse. Urban environments are a kind of mosaic and thus are attractive for beneficial entomofauna. This may contribute to the preservation of the diversity of this group of fauna. If such research is conducted at the level of anthropogenic landscape - urban and semi-natural environments, then it contributes to the development of insect ecology. I strongly support such research.

Specific Comments to update my review:

1.    The research addresses the role of ecological corridors for fauna between the urban ecosystem and the agricultural landscape.
2.    The topic is original and fills a gap in this area of research.
3.    An added novelty is the performance of surveys on a landscape scale.
4.    Research could be continued based on another trophic group, i.e. predators and/or parasitoids. In my research from the 1990s, I noticed a large abundance of aphid parasitoid populations in the urban ecosystem (Barczak 1993, habilitation thesis).
5.    The conclusions consistent with the evidence and arguments presented
and do they address the main question posed.
6.    The references are appropriate.
7.    Figures and tables sufficiently transform and illustrate the obtained data. 

Author Response

Our responses to each comment are in bold text

Research on ecological corridors for urban fauna and semi-natural habitats outside cities, including in agricultural landscapes, is extremely valuable. They have been carried out in Europe for many years. Using these corridors, both pollinators and natural enemies of pests can move - disperse. Urban environments are a kind of mosaic and thus are attractive for beneficial entomofauna. This may contribute to the preservation of the diversity of this group of fauna. If such research is conducted at the level of anthropogenic landscape - urban and semi-natural environments, then it contributes to the development of insect ecology. I strongly support such research.

Thank you so much for your support of our study!

Round 2

Reviewer 3 Report

Comments to authors.

The authors avoid assuming that preference cannot be known by studying pollen loads, particularly in oligolectic species, since it may be a simple sampling effect. Although known from the literature, pollen specialization in Asteraceae in the two species was not proven by this pollen load study, and I wonder what about the other three species? Yes, generalist bees can be temporarily specialized, but you are not sure about this and especially using this method. Section 3.2. is not indicative of preference: bees were sampled foraging directly on flowers (biased methodology as noted in previous revision). Diet breadth based on pollen load (and even including pollen from body surface) is reduced in comparison with that based on nest provisions, and can show equivocal pollen preference. The methodology used is incorrect to achieve the proposed objectives on pollen preference in bees. My proposal is the limitation of claims about preference for pollen based on an inadequate methodology to achieve it.

Author Response

Reviewer's comment in regular text and our response is in bold text

The authors avoid assuming that preference cannot be known by studying pollen loads, particularly in oligolectic species, since it may be a simple sampling effect. Although known from the literature, pollen specialization in Asteraceae in the two species was not proven by this pollen load study, and I wonder what about the other three species? Yes, generalist bees can be temporarily specialized, but you are not sure about this and especially using this method. Section 3.2. is not indicative of preference: bees were sampled foraging directly on flowers (biased methodology as noted in previous revision). Diet breadth based on pollen load (and even including pollen from body surface) is reduced in comparison with that based on nest provisions, and can show equivocal pollen preference. The methodology used is incorrect to achieve the proposed objectives on pollen preference in bees. My proposal is the limitation of claims about preference for pollen based on an inadequate methodology to achieve it."

Thank you for your comment. We believe the reviewer is requesting greater clarification in how we describe pollen preference, especially for specialist vs generalist bee species. We now add text to clarify that, like many others in the field investigating foraging patterns (Ritchie, et al 2016; Urbanowicz, et al 2017; and Alarcon, 2010) we are investigating pollen found on the body of the pollinator and not at pollen found in nest provisions. Investigation of nest provisions has classically been used to categorize a bee species as specialist or generalist (lecty, Sipes and Cane, 2006); however, our paper is focused on foraging preference and not lecty in bees. Pollen foraging preference describes which flowers bees are foraging upon given the site-level resources available to them (see Ritchie et al 2016; Redhead, et al 2016; Layak, et al 2020; and Jha et al 2013). Though we agree that an examination of both body pollen and nest pollen would be ideal to study lecty in bees, the focus of our study is not on lecty but is on the foraging preference, and our methods are consistent with many other studies that study foraging preference (de Manicor, et al 2020; Tur, et al 2014; Alcaron, 2010; Jha, et al 2013; and Ritchie, et al 2016). To clarify this distinction, we have made several changes throughout the manuscript to clarify our approach and we now use the term “pollen foraging preference” to distinguish our approach from studies focused on lecty based on pollen nest provisions.

  • First, we have changed the term “specialization” to “foraging specialization” throughout the paper to explicitly state that the networks describe foraging preferences and not lecty as our results were based on the pollen on the body and not on what pollen was found at the nests.
  • Second, we have added this sentence to the methods section (lines 157-160) to explain that building pollen-based networks by sampling pollinators directly from plants is a commonly used method. “We sampled bees systematically within our 50 x 50 m plot by collecting any native bee seen actively foraging on a flower using hand nets, as done in previous pollen studies examining foraging preferences [45–47].”
  • We also added these three references at the end of the sentence:

de Manincor, N.; Hautekèete, N.; Mazoyer, C.; Moreau, P.; Piquot, Y.; Schatz, B.; Schmitt, E.; Zélazny, M.; Massol, F. How Biased Is Our Perception of Plant-Pollinator Networks? A Comparison of Visit- and Pollen-Based Representations of the Same Networks. Acta Oecologica 2020, 105, 103551, doi:10.1016/j.actao.2020.103551.

Tur, C.; Vigalondo, B.; Trøjelsgaard, K.; Olesen, J.M.; Traveset, A. Downscaling Pollen-Transport Networks to the Level of Individuals. Journal of Animal Ecology 2014, 83, 306–317, doi:10.1111/1365-2656.12130.

Alarcón, R. Congruence between Visitation and Pollen-Transport Networks in a California Plant-Pollinator Community. Oikos 2010, 119, 35–44, doi:10.1111/j.1600-0706.2009.17694.x.

  • Finally, we added this sentence (lines 221-226) to explain why it is important to study pollen preference among both lecty-based specialist and generalist bee species. “Though Svastra petulca and Melissodes tepaneca are considered Asteraceae specialists based on the composition of pollen masses used for nest provisioning [65], nest provisions do not fully describe all floral associations by adult bees, and body pollen collected from these species has revealed that adult bees visit a variety of hosts during their foraging bouts [66]. Thus pollen-based foraging preference studies provide a great tool for understanding foraging behavior, beyond nest provision to include nectar foraging, among both specialist and generalist bee species.”